# Hepatitis B immunity and vaccine completion among adults at increased risk for hepatitis B infection in Zambia

Enock Syabbalo[1,2☯*], Sydney Mpisa[1,2,3☯], Michael J. Vinikoor[2,4,5,6], Mercy Wamundila[4], Likando Munalula[4], Taonga Musonda[4], Ruth Phiri[4], Paul Kelly[2,4,7], Chloe Thio[8], David L. Thomas[8], Edford Sinkala[1,2,4]

1 University Teaching Hospital, Lusaka, Zambia, 2 University of Zambia, Lusaka, Zambia, 3 Lusaka Apex Medical University, Lusaka, Zambia, 4 Tropical Gastroenterology and Nutrition Group, Lusaka, Zambia, 5 University of Alabama at Birmingham, Birmingham, Alabama, United States of America, 6 Centre for Infectious Disease Research in Zambia, Lusaka, Zambia, 7 Queen Mary University of London, London, United Kingdom, 8 Johns Hopkins University, Baltimore, Maryland, United States of America

☯ These authors contributed equally to this work.
* syabbaloenock@gmail.com

## Abstract

### Background

Because of its low cost and lasting effects, hepatitis B virus (HBV) vaccination of adults in Africa could significantly contribute to viral elimination. Among at-risk adult populations there are few data to inform vaccine implementation, including on pre-existing immunity and vaccine uptake. We serologically profiled adults with specific risk factors for HBV infection in urban Zambia and evaluated their uptake of vaccine.

### Methods

At a tertiary hospital in Lusaka, we recruited hepatitis B surface antigen-negative adult (age 18+) contacts to people with chronic HBV, health workers (HCWs), and people with HIV (PLWH). After we jointly collected blood and gave the first HBV vaccine dose (blind to the full serological profile), we called back those whose results showed insufficient surface antibodies (anti-HBs) to complete the 3-dose series at 1 and 6 months, and we reimbursed transport costs. Stratified by group, we described the proportions of participants with past vaccination, resolved infection (anti-HBc-positive regardless of anti-Hbs status), isolated core antibodies (anti-HBc-positive and anti-HBs-negative), and neither antibody. We described the correlates of resolved infection. We described completion of the vaccine series in anti-HBs-negatives. In those with isolated core antibodies, we explored the incidence of an anamnestic response based on post-first-dose anti-HBs > 10 IU/ml and >= 1-log increase from baseline.

**Data availability statement:** Data is available on Figshare. DOI:10.6084/m9.figshare.30000523.

**Funding:** This work was supported by the National Institutes of Health (R01AI148049 to DLT; R37AI179640 to MJV).

**Competing interests:** The authors have declared that no competing interests exist.

## Results

616 adults (median age, 32.2 years, IQR [26.7–43.8]; 61.2% women) enrolled, including 333 contacts, 213 HCWs, and 70 PLWH. Half had neither antibody, including 68.5% of health workers. Prior vaccination was seen in 8.3% overall, including 11.3% of HCWs. Resolved infection was present for 39.3% and was more prevalent with increased age and among contacts. Isolated core was present in 59 (9.6% overall and 24.7% of those with resolved infection) participants. Among the 383 participants eligible for vaccination, 377 (98.4%) received 1 dose, 190 (49.6%) received 2 doses, and 54 (14.1%) completed the series. Among 18 individuals with isolated core antibodies, none had detectable HBV DNA, and 9 (50%) had an anamnestic response.

## Discussion

Most adults at risk for HBV in Lusaka, Zambia, had inadequate immunity, which could undermine HBV elimination. High resolved infection rate in contacts supports the role of index testing in HBV case finding. Low vaccine completion, despite vaccine access and addressing transportation costs, was striking and underscores the need for integrated behavioral science approaches to improve implementation of this potentially cost-effective intervention. Low rates of anamnestic response in people with anti-HBc-positivity should be further studied in Africa.

## Introduction

With 1.2 million new infections annually [1], 300 million people chronically infected [2] and 2 billion people having been previously infected, Hepatitis B continues to be an infectious disease of public health importance globally. Viral hepatitis for which Hepatitis B forms the majority of the infections, was the 2nd leading cause of death amongst infectious diseases in 2022, behind COVID 19 [1]. Hepatitis B infection mortality is mainly due to liver cirrhosis and/or hepatocellular carcinoma. Africa is disproportionately affected by Hepatitis B, accounting for 80 million chronic cases [3] and most new global infections [4]. To reduce new infections, infant vaccination, including at birth, is a major strategy for hepatitis B elimination since infection early in life is less likely to naturally resolve than in adulthood.

Hepatitis B is potentially preventable, with vaccination being at the core of primary prevention strategies. Hepatitis B vaccination has long been considered a safe, efficacious and cost-effective method of prevention. Since the first hepatitis vaccine in 1981 [5], several iterations have been developed to optimize immunogenicity and safety [6]. The preferred site of administration is intra-deltoid and preferred vaccination schedule is over 6 months with the 2nd vaccine given 1 month after the first, and the 3rd vaccine given at 6 months although alternative schedules have been proposed. The WHO position paper of 2017 indicated that 1–2 month post vaccination hepatitis B surface (HBs) IgG antibody titres of greater than or equal to 10 mIU/ml

were considered effective to provide protection and a 3-dose strategy ensured that this target would be reached in greater than 95% of vaccinated individuals [7]. Though 10 mIU/ml is a correlate of immunity, 100 mIU/ml is considered as the optimal level of HBs and a marker of a robust immune response [8].

While infants in Africa are the priority group for vaccination, there is also strong rationale to vaccinate adults too, especially higher risk populations like health care workers, contacts of individuals infected with hepatitis, men who have sex with men (MSM), people who inject drugs (PWID), sex workers and people living with HIV (PLHIV). Data on adult transmission are lacking but do demonstrate the potential of vaccine. For example, among MSM in Kenya [9], HBV incidence based on acquisition of core antibodies was 6% per year. In Zambia, where 6% of adults have chronic infection, health care workers reported 1.3 needlestick injuries per work year [10]. Despite these data, vaccination of at-risk adult groups in Africa remains low. A systematic review reported HBV vaccination coverage among health workers in Africa of only 24.7% [11].

Low HBV vaccination in Africa, outside of infancy, may be due to limited awareness, lack of data on the need, and implementation questions. In Zambia, vaccination for hepatitis B was introduced into the national vaccination programme in 2005 with 3 doses being provided as part of a pentavalent vaccine diphtheria, tetanus, pertussis, Haemophilus influenzae type b, and hepatitis B (DPT-Hib-HepB) administered at 6, 10 and 14 weeks of age [12]. This is provided free of charge and financed by the Ministry of Health in collaboration with its cooperative partners. However, universal or targeted birth-dose vaccinations are not currently provided to reduce the risk of mother to child transmission but are being planned for future rollout as part of the 2030 HBV elimination strategy. The birth dose in addition to HBV immunoglobulins are procured at a cost that is usually prohibitive by parents identified by clinicians as at risk for transmission, further hampering elimination of mother to child transmission efforts. Free adult vaccination is not currently available although the current national health strategy advises that all at risk individuals such as health workers, contacts of people living with the infection, and immunocompromised people should be vaccinated. Vaccine is available in the private sector but reaches few adults.

One common question around HBV vaccine implementation in Africa and other low and middle-income regions with high HBV prevalence is whether to undertake serological profiling before vaccination as a proportion of individuals may already be immune from prior infection, which could reduce vaccine dose requirements. When serologically profiling in these settings, questions also arise related to isolated core antibodies, which is defined as having negative hepatitis B surface antigen (HBsAg) and surface antibodies [13,14]. In Africa, the prevalence of isolated core antibodies is much more common than in upper-income countries. In upper-income countries, these individuals are assumed to have sufficient memory T/B cell immunity and are not routinely vaccinated [15]. With vaccination, protective antibodies begin to wane over time, preventing hepatitis B for up to 20–30 years [7]. The immune system however is able produce neutralizing antibodies when exposed to previously encountered antigens through a complex interplay between differentiated B cells, CD4 and CD8 T cells [16]. However, in the African context where HBV exposure is more common, this issue needs to be better defined. The WHO does not currently recommend vaccine boosting who have successfully completed a 3-dose vaccination schedule. In the isolated anti-HBc population, several studies have shown an increase in HBs titers after 'vaccine boosting', but none have outrightly demonstrated the clinical significance of this [17,18].

Another important HBV vaccine question is how to achieve high rates of HBV vaccine series completion among at risk adults in Africa. Right now, nearly all existing multi-dose vaccination programs focus on infants. Those for adults entail campaigns that have no longitudinal component. The COVID pandemic was among the first times when adults were provided a multi-dose vaccine series in many African settings. In past efforts at HBV vaccination of adults, prohibitive cost, low level of education, being male, reduced work experience, low perceived risk and being a resident of a rural area were reported as barriers to vaccine series completion [19–21]. Providing financial incentives, making vaccines free, and occupational health promotion are recommended to overcome these barriers [22,23].

In Zambia, to address real-world gaps in the HBV prevention field, we estimated HBV immunity and the need for HBV vaccination among selected at risk adult groups including health workers, contacts to people with chronic hepatitis B

infection, and people with HIV. We expected to find high levels of prior infection and immunity among these groups. Then for non-immune individuals, we offered free vaccination with transportation reimbursement to augment completion of the 3-dose vaccine series. Therefore, the at-risk groups selected with no immunity to hepatitis B would benefit from the vaccinations provided. We expected that vaccine completion would be higher among health workers compared to other groups. In those with isolated core antibodies who received the vaccine, we assessed for the expected anamnestic response, indicating cellular immunity. The study aimed to explore immune status for hepatitis B amongst at-risk populations and provide preliminary data around vaccine uptake in these populations that could ultimately be used to direct policy and future research of hepatitis B in Zambia.

## Methods

The study recruited participants using convenience sampling from a teaching hospital in Lusaka, Zambia, as part of a multicentre study investigating the human genetics of chronic hepatitis B infection. Both verbal and written informed consent were obtained from participants, and ethical approval was obtained from the University of Zambia Biomedical Research Ethics Committee approval reference number 2604–2022. Inclusion criteria were age at least 18 years, at elevated risk for hepatitis B, and testing HBsAg-negative on a rapid diagnostic test (RDT; Determine HBsAg 2; Abbott Laboratories, USA). We focused on three groups at elevated risk according to Zambian guidelines: health workers (HCW), contacts to someone known to have chronic Hepatitis B, and people living with HIV. There were no exclusion criteria. After providing written informed consent to participate in the study, blood was collected for qualitative hepatitis B core antibody (anti-HBc) testing and qualitative and quantitative hepatitis B surface antibody (anti-HBs) testing. Participants were then offered an immediate dose of HBV vaccine with the BEVAC rDNA Hepatitis B vaccine (Biological E Limited, India), which is a second generation vaccine containing the S protein. Vaccination was not required for study participation. HIV rapid testing (Determine HIV-1/2, Abbott Laboratories, USA) was also conducted after routine counselling. After review of the serological testing results, available within 1–2 weeks, those with HBsAb < 10 mIU/ml, regardless of their core-Ab status, were phoned and asked to return for 2 additional vaccine doses at 1 month and 6 months after the first dose or to complete all 3 doses if they did not have the initial one yet. We explored vaccine immunogenicity in a subgroup of people with isolated core Ab, defined as HBsAg-negative, anti-HBc-positive, and anti-HBs < 10 IU/ml at baseline. At least one month after the first vaccine dose and before any additional doses, we repeated anti-HBs. In those without anamnestic responses (defined in the next paragraph), using stored plasma from enrollment, we checked HBV DNA viral load (Panther platform, lower limit of detection, 10 IU/ml) to rule out occult infection.

Data analysis was performed in RStudio version 2024.12.0.467 and Stata version 17 (Statacorp, College Station, TX). If a person was both HIV-positive and a contact or a HCW, we considered them in the PLWH group. We described the demographic and serological profiles by sex using Fishers exact test and Kruskal Willis Test. By group, we reported the proportion of individuals with immunity, based on anti-HBs > 10 (i.e., anti-HBs-positive) at enrollment, immunity from on vaccination, defined as anti-HBc-negative and anti-HBs-positive, and resolved infection based on anti-HBc, regardless of anti-HBs. Using multivariable logistic regression, we assessed the correlates of resolved infection including demographics and participant group. Among those with isolated core, we described the proportion with an anamnestic response, defined as at least 1-log increase and achieving anti-HBs of>=10 mIU/ml following the first dose of vaccine. Finally, after reporting the vaccine completion rate, we used multivariable logistic regression to compare the odds of vaccine completion between HCW and other groups, adjusted for other characteristics.

## Results

From October 3, 2022, to December 8, 2023, we enrolled 616 individuals, including 333 contacts to people with HBV, 213 HCWs, and 70 people with HIV. The sample also included 10 pregnant women. The median age of participants was

32.2 years (IQR, 26.7–43.8) and 377 (61.2%) were women. Overall, 234 (38.0%) had immunity based on anti-HBs. When anti-HBs was reported as non-immune (i.e., < 10 mIU/ml), the median concentration of anti-HBs was 0.99 mIU/ml (IQR, 0.40–1.00).

The distribution of HBV serological profiles varied by age and participant group (see Table 1). In brief, 323 (52.4%) were HBV-naïve, 242 (39.3%) had resolved infection, and 51 (8.3%) were previously vaccinated. In multivariable regression, when compared to HCWs, contacts (adjusted odds ratio [AOR], 2.07; 95% confidence interval [CI], 1.38–3.12)) and PLWH (AOR, 2.14; 95% CI, 0.66–7.06) had increased odds of resolved infection. Compared to participants <30 years old, older age was also associated with resolved infection, reaching 7 times increased odds at ages 50+ (Table 2). Similarly, isolated core antibodies increased with age and among contacts and PLWH, when compared to younger participants and those without HIV. Isolated core antibodies were seen in 13 (18.6%) of PLWH and 46 (8.4%) of those without HIV (P = 0.007).

Among the 59 individuals with isolated core antibodies, 18 (30.5%; 14 contacts, 1 HCW, and 3 people with HIV) underwent repeat assessment of anti-HBs levels after their first dose of vaccine. 9 of 18 (50%) had an anamnestic response based on rise in anti-HBs (see Table 3). This included 0 of 3 PLWH and 9 of 15 participants without HIV. Of the 9 non-responders, none had

**Table 1. Demographic, serological profiles, and vaccine uptake among HBsAg-negative adults eligible for HBV vaccination in Zambia.**

| | Overall (N = 616) | Population at risk for HBV | | | P value |
| --- | --- | --- | --- | --- | --- |
| | | HCWs (n = 213) | Contacts (n = 333) | PLWH (n = 70) | |
| Age, in years | | | | | |
| 18-29 | 253 (41.1) | 139 (65.3) | 108 (32.4) | 6 (8.6) | <0.001 |
| 30-39 | 155 (25.2) | 49 (23.0) | 87 (26.1) | 19 (27.1) | |
| 40-49 | 123 (20.0) | 18 (8.5) | 84 (25.2) | 21 (30.0) | |
| 50+ | 85 (13.8) | 7 (3.3) | 54 (16.2) | 24 (34.3) | |
| Sex | | | | | |
| Women | 377 (61.2) | 132 (62.0) | 191 (57.4) | 54 (77.1) | 0.008 |
| Men | 239 (38.8) | 81 (38.0) | 142 (42.6) | 16 (22.9) | |
| HBV serostatus* | | | | | |
| Prior vaccination | 51 (8.3) | 24 (11.3) | 22 (6.6) | 5 (7.1) | <0.001 |
| Resolved infection with immunity | 183 (29.7) | 41 (19.3) | 113 (33.9) | 29 (41.4) | |
| Isolated core antibodies | 59 (9.6) | 2 (0.9) | 44 (13.2) | 13 (18.6) | |
| Naïve to all HBV markers | 323 (52.4) | 146 (68.5) | 154 (46.3) | 23 (32.9) | |
| Accepted first vaccine on day of serological profiling | | | | | |
| Yes | 563 (91.4) | 197 (92.5) | 303 (91.0) | 63 (90.0) | 0.753 |
| No | 53 (8.6) | 16 (7.5) | 30 (9.0) | 7 (10.0) | |
| Vaccine doses received if deemed eligible per anti-HBs | | | | | |
| 0 doses | 6 (1.6) | 4 (2.7) | 2 (1.0) | 0 | <0.001 |
| 1 dose | 187 (48.8) | 56 (37.8) | 113 (57.1) | 18 (48.7) | |
| 2 doses | 136 (35.5) | 53 (35.8) | 69 (34.9) | 14 (37.8) | |
| 3 doses (complete series) | 54 (14.1) | 35 (23.7) | 14 (7.1) | 5 (13.5) | |

*Prior vaccination was based on negative anti-Hbc and anti-HBs >=10; resolved infection with immunity was positive anti-HBc and anti-HBs >=10; isolated core was positive anti-HBc and anti-HBs < 10.

Abbreviations: HBsAg, hepatitis B surface antigen; HBV, hepatitis B virus; HCW, healthcare worker; PLWH, person living with HIV infection.

**Table 2. Correlates of resolved HBV infection among at-risk adult populations in Zambia.**

| | Bivariable | | Multivariable* | |
| --- | --- | --- | --- | --- |
| | Odds ratio | P value | Odds ratio | P value |
| Age, in years | | | | |
| 18-29 | Reference | | Reference | |
| 30-39 | 4.10 (2.92-5.75) | <0.001 | 3.70 (2.32-5.88) | <0.001 |
| 40-49 | 4.47 (3.04-6.58) | <0.001 | 5.11 (3.10-8.44) | <0.001 |
| 50+ | 6.23 (3.88-10.00) | <0.001 | 7.45 (4.15-13.38) | <0.001 |
| Sex | | | | |
| Women | Reference | | Reference | |
| Men | 1.25 (0.96-1.63) | 0.094 | 0.92 (0.64-1.33) | 0.653 |
| Population | | | | |
| HCWs | Reference | | Reference | |
| Contacts | 3.32 (2.28-4.82) | <0.001 | 2.07 (1.38-3.12) | 0.001 |
| PLWH | 6.34 (2.03-19.76) | 0.001 | 2.14 (0.66-7.06) | 0.204 |

*Factors associated with the outcome at P<0.2 in bivariable analysis were included in the multivariable model.

Abbreviations: HBV, hepatitis B virus; HCW, healthcare worker; PLWH, person living with HIV infection.

**Table 3. First HBV vaccine dose response in participants with isolated core antibodiesantibodies.**

| ID# | Age (years) | Sex | HIV status | HBsAg | Anti-HBc | HBV DNA | Anti-HBs levels (mIU/ml) | | Anamnestic response* |
| --- | --- | --- | --- | --- | --- | --- | --- | --- | --- |
| | | | | | | | Baseline | Post-vaccine 1 | |
| R0224 | 44 | Male | Negative | Negative | Positive | --- | 0 | 445.57 | Yes |
| R0451 | 52 | Female | Negative | Negative | Positive | --- | 8.38 | 103.44 | Yes |
| R0513 | 28 | Female | Negative | Negative | Positive | --- | 0 | 30 | Yes |
| R0656 | 31 | Male | Negative | Negative | Positive | --- | 1.42 | 1000 | Yes |
| R0706 | 35 | Male | Negative | Negative | Positive | --- | 7.13 | 115.87 | Yes |
| R0821 | 60 | Female | Negative | Negative | Positive | --- | 2.37 | 33.3 | Yes |
| R0829 | 36 | Female | Negative | Negative | Positive | --- | 3.68 | 47.26 | Yes |
| R0850 | 58 | Female | Negative | Negative | Positive | --- | 3.82 | 45.04 | Yes |
| R0851 | 39 | Female | Negative | Negative | Positive | --- | 4.16 | 53.75 | Yes |
| R0486 | 47 | Female | Positive | Negative | Positive | No DNA detected | 4.7 | 10.12 | No |
| R0508 | 47 | Male | Negative | Negative | Positive | No DNA detected | 5.56 | 16.56 | No |
| R0603 | 52 | Female | Positive | Negative | Positive | No DNA detected | 8.42 | 23.73 | No |
| R0607 | 36 | Male | Negative | Negative | Positive | No DNA detected | 9.29 | 11.95 | No |
| R0635 | 66 | Female | Negative | Negative | Positive | No DNA detected | 2.37 | 3.75 | No |
| R0703 | 45 | Female | Positive | Negative | Positive | No DNA detected | 0.29 | 0.44 | No |
| R0837 | 56 | Female | Negative | Negative | Positive | No DNA detected | 0.41 | 1.62 | No |
| R0839 | 62 | Male | Negative | Negative | Positive | No DNA detected | 2.09 | 2.42 | No |
| R0852 | 34 | Female | Negative | Negative | Positive | No DNA detected | 0.14 | 1.99 | No |

* An anamnestic response was defined as at least 1-log increased and anti-HBs of >10 mIU/ml at least 1 month after first dose of hepatitis B vaccine

occult HBV based on HBV DNA testing. We did not repeat anti-HBs after additional doses of vaccine in non-responders. Further statistical analysis summarized in supplemental S1 Table found in the supplemental information shows the correlates of isolated hepatitis B core antibodies among at-risk and surface antigen-negative adult populations in Zambia.

On the day of enrollment, 565 (91.7%) agreed to receive their first dose of HBV vaccine without knowledge of their full serological profile and this was similar by group. Based on serological results from that day, we classified 383 (62.2%) participants requiring the full series at 1 and 6 months after the first dose. Because the project closed on February 28, 2024, we calculated vaccine completion rate based on eligible participants requiring vaccine who enrolled at least 8 months before project closure (i.e., June 30, 2023), and were classified as needing the full series. Among this group of 235 individuals, 229 (97.4%) had one dose, 129 (54.9%) had two doses, and 46 (19.6%) received all three doses of the vaccine. Completion of the vaccine series was similar across groups (P = 0.216) at 23.5% for HCWs, 15.4% for contacts, and 12% for PLWH. There were also no differences in vaccine completion by demographic characteristics.

## Discussion

Among adults in Lusaka, Zambia, we performed HBV serological profiling and vaccination among high-risk adults to provide evidence on the need for HBV prevention in adult populations. Concerningly, most participants had inadequate immunity. PLWH and contacts had a higher burden of resolved infection than HCWs supporting the role of index testing in HBV case finding. Among those with resolved infection, the proportion with isolated anti-HBc was relatively high, and in an exploratory analysis, only half of this group achieved an anamnestic response with vaccination, which requires further study. Despite offering transportation reimbursement, only 1 in 5 participants in the project received the 3 doses of the vaccine, highlighting the need for strong sociobehavioral strategies to accompany adult HBV vaccination efforts in the future.

One important finding of this study was that, despite hepatitis B being endemic to Zambia, many adults remain at risk of infection. We classified 40% of the sample as needing vaccination due to inadequate antibodies. This is concerning as 20–60% of at-risk groups had resolved infection, in addition to the 6% of adults living with chronic infection in Zambia based on a national population serosurvey. Strikingly few participants in this study had been vaccinated beforehand despite HBV vaccine being strongly recommended for all groups in policy documents. This included only 11% of HCWs. A larger proportion of HCWs were vaccinated compared to contacts; however, it was still inadequate. These estimates are like the Africa region overall where only ~1 in 4 HCWs is vaccinated per a meta-analysis and systematic review [11].

We also documented increased prevalence of resolved HBV infection among contacts to people with chronic HBV and among PLWH, after adjusting for age. These data provide additional evidence for transmission in families and social networks. A recent study in Democratic Republic of Congo also documented intrafamilial transmission [24], which is neglected in Africa beyond mother-to-child transmission. Our data from PLWH are also supported by several studies showing higher resolved infection in this group. Lower rates of infection HBV among HCW could be a sign of better knowledge and expertise on disease transmission and prevention. Contacts and PLWH, on the other hand, may have been less likely to have knowledge of these practices and on the modes of transmission. We also found that resolved HBV infection increased strongly with age, which was also reported in Uganda [25], and likely reflects temporal trends in behaviors and public health programs.

We also explored the issue of whether to vaccinate populations in Africa who have isolated core-Ab, which was found in ~10% of participants overall and ~20% of participants with HIV. Isolated core was unlikely to be due to a false-positive core test as modern assays have specificity of 99.88% [26]. After vaccination, only half of people with isolated core antibodies experienced the expected anamnestic response to the first vaccine dose, albeit in a small sample size (n = 18). There is a relative dearth of information related to vaccinations of populations in Africa with isolated core antibodies but this finding is supported by other studies, largely in Asia. Yao et al reported anamnestic responses in 30% of participants, and Ural et al reported it in half. On the other hand, our data conflict with Su et al who reported 92% (12 of 13) response [18] and there was a 100% anamnestic response reported in a study in Malaysia [27]. It is important to highlight that despite similarities in the high prevalence of hepatitis B and relative reduced access to testing and treatment between Africa and Asia, Africa has been identified as requiring special focus in view of having over half of new infections globally

[1]. Efforts have been made to address the disproportionate research output into hepatitis B in Africa such as through continental collaborations like HEPSANET and Africa CDC, but more research is required to explore this regional variation. Our data is an important exploration into this unique population and suggests inadequate immunity in people with isolated core in Africa and the need to vaccinate them fully. Unfortunately, we did not continue to monitor anti-HBs after the 2nd and 3rd doses. The data from this limited series also did not delve into the complex immunology surrounding vaccination in this group and should be interpreted with caution. Patients with isolated core antibodies are thought to have a hepatic reserve of hepatitis B genetic information in form of covalently closed circular and integrated HBV that may play a role in continued seroprotection in these individuals, but also pose a risk of reactivation [28]. Providing these patients with a booster may in fact provide protection from superinfection and perhaps even reactivation but more studies are needed to expand on the findings from this small cohort.

In this study, we also evaluated HBV vaccine completion in Zambia, which we tried to augment a priori by making transportation reimbursement available, avoiding stockouts, and ensuring the vaccine was routinely accessible at the hospital. Despite these recommended strategies, a disappointing 16% of participants received all 3 doses, which was significantly lower than a the ~70% in similar study on a university community from Ghana [29] and the ~60% of outpatients receiving care from 3 urban clinics in Kenya [30]. Among HCWs, uptake in Zambia was lower than Ghanian health care workers of whom ~50% completed the series [21]. Understanding reasons for low vaccine series completion were outside the scope of the project but could be related to low behavioral and social factors. Vaccine skepticism during the recent COVID pandemic may have also played a role. Yet, 91% agreed to a first dose on the day of serological profiling, suggesting high interest in being immunized. HBV elimination programs should consider other strategies to improve vaccine uptake and completion include incentivization, utilization of digital platforms and traditional media to create awareness amongst at-risk populations, increased hepatitis awareness and screening activities during commemoration days and vaccination campaigns, as well as integrating screening and hepatitis B vaccination into routine care.

This study to the best of our knowledge is one of the first in Zambia to generate data around HBV prevention in adults. Demonstrating low levels in immunity in at risk populations may push policymakers and implementers of public health strategies to address this important issue if significant progress is to be made in the elimination of hepatitis B. While our data build rationale to test and vaccinate these populations, low vaccine series completion necessitates careful planning as this is scaled-up. Our study was also one of the first studies in the region to provide information about vaccine boosting in isolated anti-HBc where guidance around this practice is not supported by rigorous evidence. These data are exploratory but might lend support to 'blind HBV vaccination' in Africa, which is providing the 3-dose vaccine series without any antibody profiling beforehand. A rapid HBsAg test is recommended however to link people to clinical care. Our data support this because even if a substantial proportion have resolved infection, they could benefit from the boost in immunity. Our data can also support a strategy focused on surface antibody testing alone, which is the correlate of protection. It is important to bear in mind the increased cost that such a strategy may have, especially where several health priorities are competing for limited funding in low resource settings. This, however, could be mitigated by the significant savings made in reduced expenditure on care for liver cirrhosis and hepatocellular carcinoma and lifelong follow up of those infected with hepatitis B. Furthermore, there is relatively finite number of adult populations at risk because of the diminishing rate of HBV infection due to the incorporation of the hepatitis 3 series vaccination in the under 5 vaccine schedule for the last 20 years. Targeting this population for immunization with a once-off national strategy may significantly reduce the prevalence of hepatitis B in the country. Other strategies that require immediate attention include universal hepatitis B birth dose administration enhanced integration in well-established services such as HIV care and antenatal care pathways where some level of hepatitis B care is present but not actively implemented or monitored as well as updating prevalence information for the country through a new demographic health survey.

While our study had strengths, including inclusion of several at risk groups, comprehensive HBV profiling, and longitudinal data on vaccine uptake and antibodies in a subgroup, it also had limitations. Limitations of the study included the small sample who had longitudinal anti-HBs measures and post vaccination serological testing on HBV naïve patients, which would

have established vaccine efficacy. However, in another study awaiting publishing of health workers in Zambia who received HBV vaccine in the absence of core antibodies, ~20% developed anti-HBs > 10 mIU/ml after the first dose of vaccination which increased to 100% after the third dose [31]. We also have limited understanding of why vaccine series completion was so low. Future qualitative research is needed in this area. A study into the immunological profile especially T and B cell immunity in the various groups studied would also be a valuable addition to the current scientific literature to better understand vaccine response. Additionally, our convenience sampling approach limits the generalizability of the study.

In conclusion, in Zambia, several populations at high risk of HBV had inadequate immunity and should be vaccinated. Vaccination efforts need to have new strategies, beyond transportation reimbursement to support vaccine series completion. Deliberate policies to enhance vaccine uptake and completion amongst at risk populations through broad-based multifaceted strategies are required. If a program is using serological profiling, vaccine boosting in isolated anti-HBc may be beneficial in improving immunity, but more studies are required to validate this strategy.

## Supporting information

**S1 Table. Correlates of isolated hepatitis B core antibodies among at-risk and surface antigen-negative adult populations in Zambia.**
(DOCX)

## Author contributions

**Conceptualization:** Enock Syabbalo, Sydney Mpisa, Michael J. Vinikoor, Ruth Phiri, Paul Kelly, Chloe Thio, David L Thomas.

**Data curation:** Enock Syabbalo, Sydney Mpisa, Michael J. Vinikoor, Mercy Wamundila, Taonga Musonda.

**Formal analysis:** Enock Syabbalo, Sydney Mpisa, Michael J. Vinikoor, Paul Kelly, Edford Sinkala.

**Funding acquisition:** Michael J. Vinikoor, Chloe Thio, David L Thomas, Edford Sinkala.

**Investigation:** Enock Syabbalo, Sydney Mpisa, Michael J. Vinikoor, Mercy Wamundila, Likando Munalula, Taonga Musonda, Ruth Phiri, Edford Sinkala.

**Methodology:** Enock Syabbalo, Sydney Mpisa, Michael J. Vinikoor, Mercy Wamundila, Likando Munalula, Taonga Musonda, Ruth Phiri, Edford Sinkala.

**Project administration:** Enock Syabbalo, Sydney Mpisa, Michael J. Vinikoor, Mercy Wamundila, Likando Munalula, Taonga Musonda, Ruth Phiri, Paul Kelly, Edford Sinkala.

**Resources:** Enock Syabbalo, Sydney Mpisa, Michael J. Vinikoor, Mercy Wamundila, Likando Munalula, Taonga Musonda, Ruth Phiri.

**Software:** Enock Syabbalo, Michael J. Vinikoor, Taonga Musonda.

**Supervision:** Enock Syabbalo, Sydney Mpisa, Michael J. Vinikoor, Likando Munalula, Taonga Musonda, Ruth Phiri, Paul Kelly, Chloe Thio, David L Thomas, Edford Sinkala.

**Validation:** Enock Syabbalo, Sydney Mpisa, Michael J. Vinikoor, Taonga Musonda, Ruth Phiri, Paul Kelly, Edford Sinkala.

**Visualization:** Enock Syabbalo, Sydney Mpisa, Michael J. Vinikoor, Paul Kelly, Edford Sinkala.

**Writing – original draft:** Enock Syabbalo, Sydney Mpisa, Michael J. Vinikoor, Paul Kelly, David L Thomas, Edford Sinkala.

**Writing – review & editing:** Enock Syabbalo, Sydney Mpisa, Michael J. Vinikoor, Paul Kelly, Chloe Thio, David L Thomas, Edford Sinkala.

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
