## [Decision Letter · Decision Letter 0]

15 Jul 2025

PONE-D-25-26849Hepatitis B immunity and vaccine completion among adults at increased risk for hepatitis B infection in ZambiaPLOS ONE

Dear Dr. Syabbalo,

Thank you for submitting your manuscript to PLOS ONE. After careful consideration, we feel that it has merit but does not fully meet PLOS ONE’s publication criteria as it currently stands. Therefore, we invite you to submit a revised version of the manuscript that addresses the points raised during the review process.

We look forward to receiving your revised manuscript.

Kind regards,

Livia Melo Villar

Academic Editor

PLOS ONE

Journal Requirements:

[This study was supported by grants from the U.S. NIH, grant number

R01AI148049 received by DLT and grant number R01AI147727 received by MJV.].

4. In the online submission form, you indicated that [Data is available on request due to ethical restrictions.].

Additional Editor Comments:

Dear Author,

Thanks for sending me this paper for my evaluation. The topic is relevant and new information was included in this manuscript. However, some issues were raised by the reviewers and they recommend the revision.

Sincerely,

Reviewers' comments:

Reviewer's Responses to Questions

**Comments to the Author**

1. Is the manuscript technically sound, and do the data support the conclusions?

Reviewer #1: Yes

Reviewer #2: Yes

Reviewer #3: Yes

2. Has the statistical analysis been performed appropriately and rigorously? 

Reviewer #1: Yes

Reviewer #2: I Don't Know

Reviewer #3: Yes

3. Have the authors made all data underlying the findings in their manuscript fully available?

Reviewer #1: Yes

Reviewer #2: Yes

Reviewer #3: No

4. Is the manuscript presented in an intelligible fashion and written in standard English?

Reviewer #1: Yes

Reviewer #2: Yes

Reviewer #3: Yes

5. Review Comments to the Author

Reviewer #1: In this manuscript, the authors describe Hepatitis B seroprevalence in adults living in Zambia and tested longitudinally the anti-HBs antibody titers detected at least 1 month after a single vaccination dose.

The study extends our knowledge of HBV incidence in African population and shows that the presence of anti-HBc antibody is associated with a rapid increased of anti-HBs titer after a single vaccination, likely caused by the presence of helper T cells specific for HBs in this population.

I have only a minor request related to the data and some general considerations about the background information and the implications of some results.

Results: Table 3 Anamnestic response

The data presented are of interest but the results lack a control since it will be important to show that individual without anti-HBc have no anamnestic response.

This review suspects that in hyperendemic area many individuals might actually have anamnestic response even in the absence of anti-Hbc antibodies, mainly due to the fact that the the accelerated level of ani-HBs production after vaccination is likely due to the existence of CD4 T cells ( specific for core or envelope) that might be present even in the absence of antibodies. In any case, the authors should show the incidence of rapid anamnestic response in individuals who are not anti-HBc+.

Having said this, the possibility that antibodies anti-HBc might not be able to define all the individuals who encounter HBV in their life time warrants discussion One of the main concern is that this manuscript focuses solely on antiviral immunity in humans as being primarily based on antibody response. However, it is evident that memory antiviral-specific immunity includes both humoral (antibody-mediated) and cellular (T cell-mediated) components. The recent COVID-19 pandemic has clearly shown that healthy individuals can contract SARS-CoV-2 without developing a robust antibody response. Nevertheless, these individuals often exhibit an anamnestic T cell response specific to SARS-CoV-2 (see for example Samandari, T. et al. Prevalence and functional profile of SARS-CoV-2 T cells in asymptomatic Kenyan adults. J. Clin. Investig. 133, (2023).). This possibility was also showed for other viral infections ( see 1.Mok, C. K. P. et al. T-cell responses to MERS coronavirus infection in people with occupational exposure to dromedary camels in Nigeria: an observational cohort study. Lancet Infect Dis 21, 385–395 (2021). 1.Zhao, J. et al. Recovery from the Middle East respiratory syndrome is associated with antibody and T cell responses. Sci Immunol 2, eaan5393 (2017).).

This reviewer is aware that the possibility that people would only mount HBV-specific T cells in the absence of anti-HBV antibodies has not been studied yet in HBV infection, and I am also not certainly asking to analyse the HBV-specific T cell response. However, the possibility that serology might not be sufficient to reveal all the past history of exposure to HBV should at least be mentioned and discussed.

There are other points that need to be properly discussed: HBV, like any other DNA virus, is never completely eliminated and as such, the lack of detection of HBV-DNA in the serum is certainly not a demonstration that HBV is not present in these individuals. We know that patients who experienced acute HBV infection still harbour HBV-DNA in their liver that is detected only by analyzing liver biopsies ( see Michalak, T. I., Pasquinelli, C., Guilhot, S. & Chisari, F. V. Hepatitis B virus persistence after recovery from acute viral hepatitis. Journal of Clinical Investigation 94, 907 (1994). Rehermann, B., Ferrari, C., Pasquinelli, C. & Chisari, F. V. The hepatitis B virus persists for decades after patients’ recovery from acute viral hepatitis despite active maintenance of a cytotoxic T-lymphocyte response. Nature Medicine 2, 1104–1108 (1996).)

This fact should be clearly discussed in the introduction. A further consequence of HBV persistence is the real benefit of vaccination in individuals who are already anti-HBc+ ( and as such likely infected by HBV and harbouring HBV-DNA). A recall vaccination might not be important in these individuals to protect them from infection since they might already carrier the virus. Vaccination in these indivduals can indeed boost anti-HBs antibody production and this might be important to help individuals better control the infection or avoid possible re-infection from different HBV strains (??) . We know that we don’t have the answer to these questions, but such a possibility should be discussed. HBV pathogenesis is complex and need to be properly represented.

Reviewer #2: This is a relevant study addressing an important public health gap in Zambia. The study provides valuable real-world data on HBV seroprevalence, isolated anti-HBc profiles, and vaccine uptake in a sub-Saharan African setting where adult HBV immunization has historically received limited attention.

To strengthen the manuscript, I recommend the following specific revisions:

Introduction:

1 - I recommend that the authors include detailed background information on Zambia’s national HBV vaccination schedule. Please specify the target age groups, whether adult vaccination is currently available and free of charge, the estimated national coverage rates (both for children and adults, if available), and the types of HBV vaccines licensed and distributed in the country. This context will help readers interpret the observed low adult vaccination rates and compare them against the vaccine introduction timeline and the age range of participants.

2 - In the second paragraph, the authors refer to a "third-generation" HBV vaccine. For clarity, especially for readers less familiar with HBV vaccine technology, I recommend adding a brief explanation of how third-generation vaccines differ from earlier formulations.

3 - Currently, the Introduction lacks details about the demographic and risk-group profile of participants. Since the Results section reveals that the cohort consisted mostly of young women (median age 32) from three specific high-risk groups (HCWs, PLWH, and contacts), I suggest adding a short paragraph in the Introduction summarizing the target population and the rationale for focusing on these groups.

Methods:

1 - In the section describing serological testing (“...blood was collected for qualitative hepatitis B core antibody (anti-HBc) testing and qualitative and quantitative hepatitis B surface antibody (anti-HBs) testing.”), please specify the exact commercial kits used for each assay, including manufacturer name, country, assay type (e.g., ELISA, chemiluminescence), and detection thresholds. This level of detail is important for reproducibility and cross-study comparability.

Results:

1 - Including participants with prior HBV exposure (resolved infection or isolated anti-HBc) in the vaccine response analysis may bias immunogenicity interpretation due to pre-existing immune memory. The authors should present stratified results for HBV-naïve individuals to allow an unbiased assessment of primary vaccine response. A dedicated figure or table focusing on this subgroup would enhance data clarity.

Discussion:

1 - The absence of serological monitoring after the second and third vaccine doses represents a limitation, as acknowledged by the authors. Without post-series serology, it is not possible to determine whether initial non-responders developed protective anti-HBs titers after completing the full vaccination schedule. I recommend that the authors more prominently highlight this limitation and discuss its implications for interpreting the vaccine efficacy data within the studied population.

Overall, this manuscript addresses a critically under-researched topic in HBV control efforts in Africa. The suggested clarifications and stratifications would further strengthen the paper and its contribution to the field.

Reviewer #3: General Comments and Commendation

I commend the authors for presenting original field data on HBV sero-status, vaccination uptake, and response among high-risk adult populations in Lusaka, Zambia. This is an important and timely contribution to the literature on hepatitis B prevention and control in sub-Saharan Africa. The manuscript is generally well-written, with well-articulated study objectives, design, and findings.

Major Review Comments for Improvement

1. Title and Abstract

The title is appropriate, and the abstract is comprehensive. However, I would suggest the following for an improved version of the abstract.

a. Review this sentence for clarity: “Low vaccine completion, despite vaccine access and transportation support, was striking, and underscores the need for integrated behavioral science approaches...”

b. The interquartile range (IQR) for median age is not specified (IQR (x, y)). Update with actual values for “x” and “y” to ensure transparency and reproducibility.

c. There is no mention of sampling method (e.g., convenience, purposive) or timeframe of recruitment. I recommend you briefly mention the recruitment method and study duration.

d. Add one comparative statement to contextualize the findings from the results.

e. Change the sub-heading, “Discussion” to “Conclusion”

2. Introduction

The introduction provides a strong foundation for this article, but I suggest you explicitly state the study objectives or significance of the study in the final paragraph of the introduction.

3. Methods

This section has a clear description of participant groups and serological testing strategy. However, it will benefit from addressing the following observed limitations:

a. The sampling technique is not abundantly clear whether it was convenience sampling or not? If it is ‘convenience sampling,’ then I suggest you address the implications for generalizability. You may also include sample size justification.

b. Provide more details about the statistical methods used. Specify the tests used for comparisons and state if there were any adjustments for confounders made for age, and sex, beyond what is currently written in this article?

4. Results

The results section provides important insights into HBV immunity gaps and low vaccine series completion in adults. The report on anamnestic response to first vaccine dose is a novel and valuable finding. However, I strongly recommend the following adjustments:

a. The use of terms such as “resolved infection,” “isolated core antibody,” and “anamnestic response” should be consistently and clearly defined earlier in the methods/results.

b. The proportion who showed anamnestic response should be accompanied by denominators and confidence intervals, where possible.

c. Summarily, it would be nice to include stratified tables for (a) immunity status by subgroup, (b) vaccine uptake by subgroup, and (c) anamnestic response among isolated core antibody group. Figure 1 already addressed the need for visual aids or bar graphs for completion rates.

5. Discussion

The discussion section impressively contextualizes the findings well. It also draws comparisons with literature from Africa and Asia, while making a strong case for policy reform and programmatic response.

However, I recommend the following areas for Improvement:

a. Even when clear sub-headings are not applied, the discussion could benefit from tighter organization addressing the following key area: Immunity Gaps, Vaccine Uptake, Anamnestic Response, and Policy Implications. Give it a thought.

b. Although the statement, “We believe these results support ‘blind HBV vaccination’ in Africa” is significant, it should be more rigorously supported with caveats. For example, cost-effectiveness and risk of unnecessary vaccination should be briefly mentioned.

c. The rationale for comparing with studies from Asia is understandable, but more attention should be given to regional differences in HBV epidemiology and health system context.

d. Summarily, I recommend expanding on implementation science considerations (e.g., what behavior change strategies might improve vaccine completion? How could digital tools or community health workers help?). Discuss how the study informs WHO guidance or Zambia’s national HBV policy.

6. Conclusion

The conclusion is sound, but it should be more specific about the next steps for research and implementation. For instance, consider recommending (1) integration of adult HBV vaccination into existing ART or ANC services, (2) operational research into vaccine adherence strategies, and (3) expanded national serosurveys.

7. References

The references are current and appropriate. It also showed a strong representation of WHO guidance and peer-reviewed African studies. However, kindly review and address the inconsistent or problematic references. For example:

a. Reference numbers 2, 4, 5, 7, 8, 9, 10, 14,15, 16, 17, 18, 19, 24 (check the authors initials), 27 (incomplete author list, crosscheck the source to be sure), and 30, either lack full citation details, have capitalization concerns, missing volume issue or pages, or has concerns with authors’ initials.

b. Crosscheck, it looks like reference number 10 is repetitive or similar to reference number 1.

c. Crosscheck, it looks like reference number 21 is repetitive or similar to reference number 13.

8. General Minor Comments

a. For line-editing to improve grammar and flow, consider changing “lower rates of infection HBV among HCWs” to “lower HBV infection rates among HCWs”.

b. Consider replacing non-academic phrases like “strikingly few” with more measured terms like “a low proportion.”

c. I suggest that you clarify in the discussion whether “blind vaccination” would apply only to high-risk adults or to general adult populations.

d. Review and ensure that acronyms are consistently defined at first use (e.g., PLWH, HCW, anti-HBc, HBsAg).

e. I suggest you specify if this was a cross-sectional or longitudinal study, especially as it seems like the language is occasionally inconsistent.

e. Take some time to read through the entire work again and carry out some editing and rephrasing, to enhance clarity and flow of the article.

Well done to the authors!!!

6. PLOS authors have the option to publish the peer review history of their article (what does this mean? ). If published, this will include your full peer review and any attached files.

**Do you want your identity to be public for this peer review?** For information about this choice, including consent withdrawal, please see our Privacy Policy .

Reviewer #1: **Yes:** Antonio Bertoletti

Reviewer #2: No

Reviewer #3: **Yes:** Ojore Godday Aghedo

---

## [Author Response · Author response to Decision Letter 1]

1 Sep 2025

The response to reviewers has been attached as well as the manuscript and revised manuscript with tracked changes.

---

## [Decision Letter · Decision Letter 1]

23 Oct 2025

PONE-D-25-26849R1Hepatitis B immunity and vaccine completion among adults at increased risk for hepatitis B infection in ZambiaPLOS ONE

Dear Dr. Syabbalo,

Thank you for submitting your manuscript to PLOS ONE. After careful consideration, we feel that it has merit but does not fully meet PLOS ONE’s publication criteria as it currently stands. Therefore, we invite you to submit a revised version of the manuscript that addresses the points raised during the review process.

We look forward to receiving your revised manuscript.

Kind regards,

Livia Melo Villar

Academic Editor

PLOS ONE

Journal Requirements:

Additional Editor Comments:

Dear Author,

i have read the paper and comments of the reviewers, so I suggest to make minor corrections,

best

Livia

Reviewers' comments:

Reviewer's Responses to Questions

**Comments to the Author**

1. If the authors have adequately addressed your comments raised in a previous round of review and you feel that this manuscript is now acceptable for publication, you may indicate that here to bypass the “Comments to the Author” section, enter your conflict of interest statement in the “Confidential to Editor” section, and submit your "Accept" recommendation.

Reviewer #1: (No Response)

Reviewer #2: All comments have been addressed

Reviewer #3: All comments have been addressed

2. Is the manuscript technically sound, and do the data support the conclusions?

Reviewer #1: Yes

Reviewer #2: Yes

Reviewer #3: Yes

3. Has the statistical analysis been performed appropriately and rigorously? 

Reviewer #1: I Don't Know

Reviewer #2: Yes

Reviewer #3: Yes

4. Have the authors made all data underlying the findings in their manuscript fully available?

Reviewer #1: Yes

Reviewer #2: Yes

Reviewer #3: Yes

5. Is the manuscript presented in an intelligible fashion and written in standard English?

Reviewer #1: Yes

Reviewer #2: Yes

Reviewer #3: Yes

6. Review Comments to the Author

Reviewer #1: The authors have partially addressed some of my specific questions and provided additional information. However, there remains a lack of detail and clarity in the abstract and throughout the manuscript.

Since this is a paper about HBV vaccination, the authors should at least include information about the composition of the HBV vaccine used. The issue is that in their new introduction, the authors wrote that “Since the first hepatitis vaccine in

1981[5], several iterations have been developed to optimise immunogenicity and

safety. The third (current) generation of hepatitis B vaccines are recombinant,

containing multiple additional surface antigens (preS1, pre-S2 and S), whereas earlier

vaccines had only contained S antigens, which can provide a more robust immune

response even from non-responders [6].”

The authors seem to suggest that their vaccine now contains the Large envelope protein (PreS1, PreS2 and S), something that I sincerely doubt. Currently, the most widely used vaccine worldwide contains only S, not PreS1 and PreS2.

PreS1/PreS2/S vaccines have been developed; they are highly immunogenic and do not require the full three-dose course, but they are also more expensive (this vaccine needs to be produced in mammalian cells). The authors used the BEVAC rDNA Hepatitis B vaccine (Biological E Limited, India), but I am unable to determine whether this vaccine contains only S or also PreS1/PreS2/S. While this information might not be essential for writing a report about “vaccine acceptance in the population,” since the authors claim that “full course of HBV vaccination should be implemented in individuals with positivity of anti-HBc, " it is important to understand exactly which vaccine was used and which adjuvant it contained.

Additionally, because the analysis of the vaccine's immunogenicity in anti-Hbc+ individuals was conducted in only 18 patients, the data are at best preliminary and should not be presented as a major result of this report.

Reviewer #2: The authors have satisfactorily addressed the revisions requested during my first review and have provided clear clarifications to the points previously raised. I now consider the manuscript appropriate for publication.

Reviewer #3: (No Response)

7. PLOS authors have the option to publish the peer review history of their article (what does this mean? ). If published, this will include your full peer review and any attached files.

**Do you want your identity to be public for this peer review?** For information about this choice, including consent withdrawal, please see our Privacy Policy .

Reviewer #1: No

Reviewer #2: No

Reviewer #3: **Yes:** Ojore Godday Aghedo

---

## [Author Response · Author response to Decision Letter 2]

11 Nov 2025

The vaccine used in the study was indeed a 2nd generation hepatitis b vaccine and as such contained the S antigen as it is the most widespread. This has been added to the introduction as well as the methods section.

A caveat has been made in the discussion section to further highlight the preliminary nature of the results shown from the anti HBc+ individuals.

---

## [Decision Letter · Decision Letter 2]

10 Dec 2025

Hepatitis B immunity and vaccine completion among adults at increased risk for hepatitis B infection in Zambia

PONE-D-25-26849R2

Dear Dr. Syabbalo,

We’re pleased to inform you that your manuscript has been judged scientifically suitable for publication and will be formally accepted for publication once it meets all outstanding technical requirements.

Kind regards,

Livia Melo Villar

Academic Editor

PLOS One

Additional Editor Comments (optional):

Dear Author,

After reading the comments of the reviewers, I recommend the publication of this manuscipt,

sincerely

Livia

Reviewers' comments:

Reviewer's Responses to Questions

**Comments to the Author**

1. If the authors have adequately addressed your comments raised in a previous round of review and you feel that this manuscript is now acceptable for publication, you may indicate that here to bypass the “Comments to the Author” section, enter your conflict of interest statement in the “Confidential to Editor” section, and submit your "Accept" recommendation.

Reviewer #1: All comments have been addressed

Reviewer #3: All comments have been addressed

2. Is the manuscript technically sound, and do the data support the conclusions?

Reviewer #1: Yes

Reviewer #3: Yes

3. Has the statistical analysis been performed appropriately and rigorously? 

Reviewer #1: Yes

Reviewer #3: Yes

4. Have the authors made all data underlying the findings in their manuscript fully available?

Reviewer #1: Yes

Reviewer #3: Yes

5. Is the manuscript presented in an intelligible fashion and written in standard English?

Reviewer #1: Yes

Reviewer #3: Yes

6. Review Comments to the Author

Reviewer #1: (No Response)

Reviewer #3: (No Response)

7. PLOS authors have the option to publish the peer review history of their article (what does this mean? ). If published, this will include your full peer review and any attached files.

**Do you want your identity to be public for this peer review?** For information about this choice, including consent withdrawal, please see our Privacy Policy .

Reviewer #1: No

Reviewer #3: **Yes:** Ojore Godday Aghedo

---

## [Editor Report · Acceptance letter]

PONE-D-25-26849R2

PLOS One

Dear Dr. Syabbalo,

I'm pleased to inform you that your manuscript has been deemed suitable for publication in PLOS One. Congratulations! Your manuscript is now being handed over to our production team.

Kind regards,

on behalf of

Dr. Livia Melo Villar

Academic Editor

PLOS One